# Hollow Forged AHD Steel Rotor Shafts for Wind Turbines – A Case Study on Power Density, Costs and GWP

Christian Hollas<sup>1</sup>, Georg Jacobs<sup>1</sup>, Vitali Züch<sup>1</sup>, Julian Röder<sup>1</sup>, Moritz Gouverneur<sup>2</sup>, Niklas Reinisch<sup>2</sup>, David Bailly<sup>2</sup>, Alexander Gramlich<sup>3</sup>

<sup>5</sup> <sup>1</sup>Chair for Wind Power Drives, RWTH Aachen University, Aachen, 52074, Germany <sup>2</sup>Institute of Metal Forming, RWTH Aachen University, Aachen, 52074, Germany <sup>3</sup>Steel Institute, RWTH Aachen University, Aachen, 52074, Germany

Correspondence to: Christian Hollas (christian.hollas@cwd.rwth-aachen.de)

Abstract. Hollow forging and air hardening ductile (AHD) forging steels are a novel manufacturing process and steel grade for the wind energy sector. Together they enable new rotor shaft design possibilities for wind turbines. Hollow forging combines the high material strength of a solid forged shaft with direct inner contour manufacturing similar to casting. To compare an AHD steel hollow forged rotor shaft to a state-of-the-art cast rotor shaft, a case study is carried out, focusing on power density, manufacturing costs and (manufacturing) global warming potential (GWP). To ensure comparability between the hollow forged and cast rotor shaft, two predesigns of a main bearing unit (MBU, rotor shaft, main bearings, bearing

15 housings) are generated via a structural integrity assessment and calculation of the bearing lifetime according to ISO 76 / 281. The resulting hollow rotor shaft has 37 % less mass than the cast rotor shaft, corresponding to a 16.5 % lower MBU mass. For the hollow forged rotor shaft to be comparable to casting regarding manufacturing costs, the forging surcharges need to be greatly reduced. Due to the shortened heat treatment of AHD steels and the use of green steel, the GWP of hollow forging is comparable to casting.

# 20 1 Introduction

To reduce the levelized cost of electricity (LCOE) of wind energy, there has been a notable, still ongoing shift towards larger, wind turbines (WT) with higher power ratings (Nejad et al., 2022). The growth in power and rotor diameter directly increase the loads on the WT drive train (Euler et al., 2023). To cope with the increased loads, the drive train needs to be redesigned resulting in heavier components. Consequently, the costs of the drive train components rise, while the required, sturdier tower

and increased logistic expenditure further add to the CAPEX (<u>cap</u>ital <u>expenditure</u>). Therefore, a need for an increase in drive train power density (power per mass) arises.

Taking the rotor shaft as an example, via novel manufacturing processes like hollow forging (Kwon et al., 2016), combined with new steel grades like air hardening ductile (AHD) forging steels, a leap in main bearing unit (MBU) power density is possible. The increase in power density is a result of better material utilization and higher material strengths compared to state

of the art manufacturing processes and materials. AHD steels harden directly during the cooling from the forging heat, resulting

in a martensitic microstructure, similar to standard high strength steels. Thus, the energy intensive quench and tempering (QT) process is omitted, reducing energy consumption and therefore global warming potential (GWP) compared to QT forging steel (Gramlich et al., 2023). As shown in a previous study (Hollas et al., 2024), MBUs with hollow forged rotor shafts achieve higher power densities than those with cast rotor shafts, particularly when forged from AHD steel.

- A hollow forged shaft is formed over a cylindrical or conical mandrel, which defines the inner geometry of the shaft. Hollow forging combines the high material strengths of QT steel (solid forging) with the direct manufacturing of the shaft bore (similar to casting). A larger inner shaft diameter (d) enables higher section modulus (S<sub>Bending</sub>, see Eq. (1)) for a fixed shaft cross-section (A; with outer shaft diameter D). This results in better material utilization regarding bending moments, which are the dominating loads on WT rotor shafts (Manwell et al., 2009; Euler et al., 2023), thus increasing power density. Solid forged
- rotor shafts must be bored to access the pitch system in the rotating hub, a. o. resulting in additional process costs and material waste. To reduce manufacturing costs, their inner diameter is kept smaller, decreasing their power density.

$$S_{Bending} = \frac{\pi}{32} \cdot \frac{(D^4 - d^4)}{D} \tag{1}$$

$$A = \frac{\pi}{4} \cdot \left(D^2 - d^2\right) \tag{2}$$

Whether hollow forged rotor shafts are economically advantageous or if the air hardening ability of AHD steel makes the global warming potential of a cast and hollow forged shaft comparable remains unanswered. Hence, this work conducts a case study to investigate this matter. The case study predesigns a MBU with an AHD steel hollow forged rotor shaft for a known

2.3 MW base load-optimised wind turbine and compares it with a predesigned cast shaft MBU design regarding power density, manufacturing costs and GWP.

# 2 Method

The case study in this work examines power density, manufacturing costs and GWP:

- <u>Power density:</u> Two new MBU variants with a hollow forged and cast rotor shaft are predesigned for the same exemplary WT to ensure comparability. Rotor shaft designs are heavily dependent on the main bearing diameter, so the entire MBU is predesigned to get the optimal power densest rotor shaft and main bearing combination. Open source WT predesign tools like NREL's *DrivetrainSE* (implemented into *WISDEM*) (National Renewable Energy Laboratory, 2024; Guo et al., 2015) cannot consider hollow forged rotor shafts. Therefore, an MBU predesign tool presented in a prior publication (Hollas et al., 2024) is used. The tool maximises the MBU power density by adjusting the rotor shaft geometry and main bearings while
- keeping the shaft length and flange size fixed. By comparing the cast and hollow forged MBU variants, the potential in power density increase can be quantified. Other MBU components are either simplified (bearing housings, assembly components) or omitted (machine carrier) in this study given their complex geometry.

The <u>manufacturing costs</u> of the MBU are estimated based on available literature such as an *onshore wind turbine CAPEX estimation model* by REICHARTZ ET AL. (2024) or *early cost estimations for forged parts* by KNIGHT (1992). The component

costs are calculated solely based on their mass or material requirement. As cost estimations for hollow forged shafts are not published, a cost model of solid cast parts was adjusted for this study. Due to cartelisation rules, most cost parameters are not publicly available and have to be estimated, such as material prices (cf. Gramlich et al., 2024).

The <u>GWP</u> of the cast and hollow forged rotor shafts are analysed, using the carbon footprint calculator *FRED* (FRED GmbH, 2024) and industry feedback. The GWP of the rotor shafts are a result of raw material emissions (material level), manufacturing and machining emissions as well as transportation emissions (process level, cf. Hagedorn et al., 2022). Each manufacturing

and machining emissions as well as transportation emissions (process level, cf. Hagedorn et al., 2022). Each manufacture step is linked to sector-specific average emission values, depending on the energy sources used.

# 3 Case study

#### 3.1 Exemplary wind turbine

This case study uses the base load-optimised WT *maxcap141* as an example, made by the engineering service provider
windwise GmbH (2024). The WT was tested in cooperation with the Chair for Wind Power Drives (Krause, 2024), providing deep insight into the MBU design and loads for this case study. The WT uses a four-point suspension in a locating/non-locating

- bearing arrangement. The maximum fatigue and extreme shaft load case during WT operation acting on the rotor shaft hub flange are given in Table 1. The extreme load case represents the critical load combination, dominated by the high bending moments of the rotor, while multiple fatigue load series were converted into one damage equivalent fatigue load case.
- Table 1: Fatigue and extreme load case of the rotor shaft

|                                                                                                             | Thrust Force                                                                                         | Shear Force         | Shear Force         | Torque Moment        | Bending Moment       | Bending Moment       |  |
|-------------------------------------------------------------------------------------------------------------|------------------------------------------------------------------------------------------------------|---------------------|---------------------|----------------------|----------------------|----------------------|--|
|                                                                                                             | F <sub>x</sub> [kN]                                                                                  | F <sub>y</sub> [kN] | F <sub>z</sub> [kN] | M <sub>x</sub> [kNM] | M <sub>y</sub> [kNM] | M <sub>z</sub> [kNM] |  |
| Fatigue Loads*                                                                                              | 450                                                                                                  | 0                   | -3150               | 2050                 | 9900                 | 0                    |  |
| Extreme Loads 350 50 -1600 2700 -12400 -1000                                                                |                                                                                                      |                     |                     |                      |                      | -1000                |  |
| Coordinate System: x-axis along shaft axis toward gearbox, z-axis opposite direction of gravity, see Fig. 1 |                                                                                                      |                     |                     |                      |                      |                      |  |
| * Da                                                                                                        | * Damage equivalent loads interpolated for a gradient of 10.8 and a reference cycle number of $10^6$ |                     |                     |                      |                      |                      |  |

# 3.2 MBU designs and power density analysis

The original MBU comprises a cast rotor shaft with a mass of 16.2 t, a non-locating 2.1 t cylindrical roller bearing on the rotor side (upwind) and locating 0.9 t double-rowed tapered roller bearing on the gearbox side (downwind). To fasten the main bearings, assembly components with a total mass of 2.5 t are used. The bearing housings are integrated into the cast machine

- frame and connect both main bearings and the gearbox to the azimuth bearing of the nacelle (see Fig. 1). The cast rotor shaft is made from EN-GJS-400-18 via a sand-casting process. The biggest wall thickness of the finished shaft is above 300 mm and therefore close to the technically feasible limit of iron casting ( $\approx 400$  mm). Larger wall thicknesses increase the risks of material defects, possibly resulting in unusable parts. The use of permanent mould casting allows for a higher cooling rate, resulting in better material properties due to a finer grain structure. This enables larger wall thicknesses but is only used for
- large series production due to the high metal mold costs. Sand casting requires a surface surcharge of up to 50 mm,

approximately 20 % of the shaft mass, and 10 % additional material for sprue and other material loses (Weiß, R., 2024). Therefore, the material input is estimated to be 21.3 t of cast iron.

(with integrated main bearing housings)

# Figure 1: Sketch of the maxcap141 MBU in a half section view and adjacent components in full view.

- For the case study comparing cast and AHD steel hollow forge rotor shafts, two variants of the *maxcap141* MBU are generated through the aforementioned MBU predesign tool in *Matlab* (The MathWorks, Inc., 2023). This ensures that the design of both MBU variants has the same level of detail and underlaying assumptions (e. g. loads, boundary conditions). The inputs include the fixed geometry of the original shaft (shaft length, flange diameter and thickness, hub flange screw connection) and shaft loads among others. Both manufacturing processes induce restrictions on the shaft design. For example, casting can produce
- conical shaft segments, while hollow forging can only produce cylindrical shaft segments with a maximum diameter jump of 600 mm between segments.

AHD steels are standardized and commercialized with a focus on small and medium die forgings (material number 1.5132). To increase the hardenability for larger wall thicknesses, modifications were developed alongside this study and the resulting alloy is currently under thorough investigation regarding the tensile properties, Charpy V-notch toughness as well as fatigue

resistance. Preliminary results show that boron, chromium and nickel can be used to further increase the air-hardening potential of AHD steels, enabling the production of components with wall thicknesses larger than 300 mm. The static material strength of the used AHD steel alloy and cast iron can be found in Table 2 in comparison to the standard QT forging steel 42CrMo4. The fatigue strengths are derived from the static material properties via the German FKM-Guideline (cf. Forschungskuratorium Maschinenbau, 2020), including the reduction in material strength over the wall thicknesses. While the present study uses the