# Peer review of "Hollow Forged AHD Steel Rotor Shafts for Wind Turbines – A Case Study on Power Density, Costs and GWP"

_Wind Energy Science, 2025_

## Referee Comment (RC1)

In this paper, the authors describe a comparison of the main bearing unit for two different versions in a 2.3 MW wind turbine: one with a standard cast iron shaft and the other with an air hardened ductile (AHD) shaft. They then compare the masses, costs and global warming potential of each variant. In general the article is well written and quite detailed. Having said that, I offer the following comments for consideration to improve the article, in many cases for the more general wind energy reader who might not be very familiar with steel processing.

**Abstract**

- Line 18: Instead of the somewhat vague "greatly reduced" I think it would be better to state the figure from the Conclusions such as "reduced by X%". Please see later comment on the Conclusions though regarding the exact percentage.
- Line 19: As noted in the Conclusions, I think it would be beneficial to the reader to add the caveat that "the GWP of hollow forging is comparable to casting and each are actually less than one-third of the GWP of the entire MBU."

**1 Introduction**

• Line 37: I am not sure I understand the explanation here regarding a higher inner shaft diameter, section modulus, area, and outer shaft diameter. A fixed section area A and outer diameter D means a fixed inner shaft diameter d as well. In looking at Figures 2 and 3, it appears the outer diameters are similar (1000 mm), but the inner diameter and section area of the AHD shaft in Figure 2 is smaller than the case diameter shaft in Figure 3.

**3.2 MBU designs and power density analysis**

- I believe the main points of this section are to summarize the design characteristics of the original maxcap 141 MBU, then the hollow forged shaft MBU, then the cast shaft MBU. I think the main point of including the maxcap 141 characteristics is to show that the other designed MBUs discussed in the remainder of the paper are "similar enough" to the actual maxcap 141. Having said that, I'll admit that I got a bit lost by the relatively long text explanations. A summary mass Table I think would be really helpful and shorten some of the text, especially as related to lines 145-162. Tables are used elsewhere in the article to good effect.
- Line 80 and Figure 1: Terminology in this sentence, the figure, and the remainder of the paper are not quite the same, leading to a little bit of confusion. I believe it would be clearer to label both the "Upwind main bearing" and "Downwind main bearing" rather than just one "Main bearing" in the figure. Are the assembly components the same as the red components? Also, the "Machine frame" is labeled twice for both the green and grey components, with the green component (I think) called the "Main bearing housing" in the remainder of the paper (for example, in Section 3.3.3 and Table 5) and the grey component separate and not in the scope of the study. Additionally, the text states "The bearing housings...connect...gearbox to the azimuth bearing". I don't believe this is correct: as far as I can tell from the figure the gearbox is only connected to the grey "Machine frame", but not the "Main bearing housing". But I could be wrong here. It does not seem to be worth mentioning the azimuth bearing, as it is not shown in the figure and not in the scope of the paper either.
- Line 85: Not being an expert in this area, I am not familiar with the term "...requires a surface surcharge...". Can this be explained a bit? Does this effectively mean a "...requires a higher wall

thickness..."? It is used throughout the paper, so a short statement here to its meaning would be beneficial.

**3.3.1 (Hollow) forging manufacturing costs**

• Lines 199-203: Is it a correct understanding that the cost of the drilling of the solid forged shaft is not included in the analysis?

**3.3.2 Casting manufacturing costs**

• Line 207 and Equation 6: It's minor, but is there a reason with the mass is denoted with subscript RS,C compared to Shaft in Equations 4 and 5?

**3.3.5 Manufacturing cost comparison**

• Line 242: Again, not being an expert in this area, I am not familiar with the term "...axial upsetting". Can a brief description be added here? Does this refer to standing the shaft upright, such that gravity has some compressive effect on it?

**4 Conclusions and Outlook**

- Line 350: Here I am not clear on the percentages "total forging surcharges need to be reduced to around 50%". Does this literally mean in line 347 that instead of "160% in total compared to 32% for casting" it would have to be "50% in total compared to 32% for casting" to have similar costs, or would it be "80% in total compared to 32% for casting". I hope this question makes sense it is a matter of discussing percentages of percentages. I think it is the former, thus when discussing the surcharge it has to be reduced from 160% to 50% for similar costs.
- Lines 352 onward: Maybe missing in the discussion of the GWP is that both shafts contribute less than 1/3 of the GWP of the total MBU. A larger portion actually comes from the bearings and housing. I wouldn't have necessarily expected this.
- Figure 6: In the middle figure, what does "Rotor shaft (lower)" and "Rotor shaft (spread)" mean? I don't believe I've seen this explained yet. Does this indicate the lower bounds and upper bounds for the shaft (i.e. the spread)?

**Minor grammatical comments:**

- Line 14: I think it would be clearer to say "...main bearing unit (MBU), consisting of the rotor shaft, main bearings, and bearing housings, are..."
- Line 18: I think it would be clearer to add "...the manufacturing GWP of hollow forging..."
- Line 172: Although it's October and near Halloween, "costumer" should be "customer" here 2.
- Line 181: Rather than "calculates to 2", is this more accurately "assumed to be 2"? Or was this truly "calculated" somehow?
- Line 230: Rather than "economically", I think "economical" is better.
- Line 258: A space is needed for "fromTable 6".

---

## Referee Comment (RC2)

**Referee comment for wes-2025-94**

Title: Hollow Forged AHD Steel Rotor Shafts for Wind Turbines – A Case Study

on Power Density, Costs and GWP

Author(s): Christian Hollas et al.

MS No.: wes-2025-94

MS type: Research article

**Objective and Strengths**

**Objective**

The study investigates the feasibility and performance of using hollow forged rotor shafts made from **Air Hardening Ductile (AHD) steel in** wind turbine **main bearing units (MBUs)**, **comparing** them to **conventional cast iron shafts**. The focus is on:

- 1. Power density
- 2. Manufacturing costs
- 3. Global Warming Potential (GWP)

**Strengths**

**1. Relevant Topic**

The paper addresses a critical challenge in wind turbine drivetrain design: increasing power density while reducing environmental impact—aligned with industry trends and world climate goals.

**2. Comprehensive Methodology**

The integration of structural analysis, cost modeling, and life cycle GWP assessment provides a holistic view, enhances by the use of real turbine data (maxcap141).

**3. Clear Engineering Insight**

The explanation of how hollow forging improves section modulus and material utilization is technically sound and well-illustrated.

**4. Transparency and Reproducibility**

The authors provide detailed assumptions, equations, and even offer to share the predesign tool, which supports reproducibility.

**Clarifications needed**

**(1) Line 53: **DrivetrainSE** does not consider hollow forged rotor shafts**

**The Claim in Question**

In the paper, the authors state:

"Open source WT predesign tools like NREL's DrivetrainSE (implemented into WISDEM) cannot consider hollow forged rotor shafts."

This suggests that DrivetrainSE lacks the capability to model hollow forged shafts — but this is **misleading** or at least **oversimplified**.

**What DrivetrainSE Can Actually Do**

**DrivetrainSE**, as part of NREL's **WISDEM** framework, **does support modeling hollow shafts**. Specifically:

- Hollow Shaft Geometry: DrivetrainSE allows users to define outer and inner diameters of the main shaft, enabling the modeling of hollow geometries.
- Mass and Inertia Calculations: It computes mass properties, stiffness and modulus based on these geometric inputs.

 Material Properties: Users can specify different materials, including highstrength steels, though not necessarily AHD steel by default.

**Evidence from cited 'Hollas et al. (2024)'**

In same authors' earlier 2024 paper, they used DrivetrainSE to benchmark their custom MBU predesign tool. They acknowledged that DrivetrainSE could model **hollow shafts**, but not the **specific manufacturing constraints of hollow forging** — such as:

- Maximum diameter jumps between shaft segments
- Forging surcharges and material flow constraints
- Air-hardening behavior of AHD steel

**Clarifying the Distinction**

So, the **correct interpretation** is:

- DrivetrainSE can model hollow shafts geometrically and structurally.
- It cannot model the manufacturing constraints and process-specific limitations of hollow forging, such as those relevant to AHD steel.

This nuance is important. The present paper could have been clearer, avoiding confusion and improving technical accuracy and transparency, by stating:

"DrivetrainSE does not natively support the manufacturing constraints and material behavior specific to hollow forged AHD steel shafts."

**(2) Sec.3.2: Bearing usage inconsistency: CRB-TRB or SRBs?**

There seems to be a **terminological inconsistency** in the paper that could be considered an error or at least a point needing clarification.

**Comparison of the 2 Statements about Bearing usage Section 3.2, First Sentence:**

"The original MBU comprises a cast rotor shaft with a mass of 16.2 t, a non-locating 2.1 t cylindrical roller bearing on the rotor side (upwind) and locating 0.9 t double-rowed tapered roller bearing on the gearbox side (downwind).

This clearly specifies:

- CRB (non-locating, upwind)
- TRB (locating, downwind)

**Line 155:**

"Both variants have similar bearing configurations, sharing the downwind spherical roller bearing and using a comparable spherical roller bearing upwind with a 60 mm inner diameter difference"

This contradicts the earlier statement by referring to **SRBs** on both sides.

**Technical Implication**

- CRBs and TRBs are fundamentally different from SRBs in geometry, load capacity, and misalignment tolerance — and the authors do have knowledge of this.
- The choice of bearing type affects:
  - Load distribution
  - Shaft deflection
  - Fatique life
  - Assembly and alignment strategies

If the design truly uses CRB and TRB, then referring to SRBs later is incorrect — unless the predesigned variants differ from the original configuration and this change was not clearly stated.

**Suggested Clarification**

It is recommended for the authors clarify:

• Whether the **bearing types were changed** in the predesign variants.

- If so, why SRBs were selected instead of CRB/TRB especially given their different stiffness and misalignment behavior.
- If not, then line 155 should be corrected to match the original bearing specification.

**Improvement Suggestions**

**1. Material Property Uncertainty**

The fatigue behavior of AHD steel is extrapolated using FKM guidelines, which are not validated for this alloy (line 105). The paper acknowledges this but does not quantify the impact on reliability or safety margins.

**2. Lack of Reliability Analysis**

The paper addresses structural integrity well, however there is **no formal reliability-based design or probabilistic treatment of uncertainties** (uncertainty quantification; e.g., in loads, material properties, forging tolerances). This limits confidence in the robustness of the design.

**3. Economic Viability Discussion**

The cost model is insightful but heavily reliant on assumed surcharges and outdated references (eg. Knight, 1992). A sensitivity analysis on cost drivers (e.g., alloy price, forging complexity) would strengthen the conclusions.

**4. Digital Twin or Monitoring Integration**

Given the trend toward condition-based maintenance, the study could benefit from discussing how hollow forged shafts might affect monitoring strategies or digital twin integration.